Is Xenopus laevis introduction linked with Ranavirus incursion, persistence and spread in Chile?

Peñafiel-Ricaurte Alexandra alexandra.penafiel.r@gmail.com 1 2
Price Stephen J. 3
Leung William T.M. 2
Alvarado-Rybak Mario 1 2 5
Espinoza-Zambrano Andrés 4
Valdivia Catalina 1
Cunningham Andrew A. 2
Azat Claudio claudio.azat@unab.cl 1
1 Sustainability Research Centre & PhD in Conservation Medicine Program, Life Sciences Faculty, Universidad Andres Bello , Santiago , Chile
2 Institute of Zoology, Zoological Society of London , London , United Kingdom
3 UCL Genetic Institute , London , United Kingdom
4 Escuela de Medicina Veterinaria, Facultad de Ciencias de la Vida, Universidad Andrés Bello , Santiago , Chile
5 Núcleo de Ciencias Aplicadas en Ciencias Veterinarias y Agronómicas, Facultad de Medicina Veterinaria y Agronomía, Universidad de las Américas , Santiago , Chile
Measey John
Electronic publication date: 2023 Feb 27
Publication date: 2023
Volume: 11
Electronic Location ID: e14497
Received 2022 Jun 22; Accepted 2022 Nov 10
Copyright: ©2023 Peñafiel-Ricaurte et al.
Copyright year: 2023
Copyright holder: Peñafiel-Ricaurte et al.
License: This is an open access article distributed under the terms of the Creative Commons Attribution License, which permits unrestricted use, distribution, reproduction and adaptation in any medium and for any purpose provided that it is properly attributed. For attribution, the original author(s), title, publication source (PeerJ) and either DOI or URL of the article must be cited.
License URL: https://creativecommons.org/licenses/by/4.0/

Keywords: African clawed frog, Ranavirus, Frog Virus 3, Reservoir, Amphibians, Emerging infectious disease

Funding: Institute of Zoology, Zoological Society of London (ZSL) Chilean National Science and Technology Fund FONDECYT N° 1211587 This research was funded by the Institute of Zoology, Zoological Society of London (ZSL), and the Chilean National Science and Technology Fund (FONDECYT N° 1211587). The funders had no role in study design, data collection and analysis, decision to publish, or preparation of the manuscript.

==============================
Ranaviruses have been associated with amphibian, fish and reptile mortality events worldwide and with amphibian population declines in parts of Europe. Xenopus laevis is a widespread invasive amphibian species in Chile. Recently, Frog virus 3 (FV3), the type species of the Ranavirus genus, was detected in two wild populations of this frog near Santiago in Chile, however, the extent of ranavirus infection in this country remains unknown. To obtain more information about the origin of ranavirus in Chile, its distribution, species affected, and the role of invasive amphibians and freshwater fish in the epidemiology of ranavirus, a surveillance study comprising wild and farmed amphibians and wild fish over a large latitudinal gradient (2,500 km) was carried out in 2015–2017. In total, 1,752 amphibians and 496 fish were tested using a ranavirus-specific qPCR assay, and positive samples were analyzed for virus characterization through whole genome sequencing of viral DNA obtained from infected tissue. Ranavirus was detected at low viral loads in nine of 1,011 X. laevis from four populations in central Chile. No other amphibian or fish species tested were positive for ranavirus, suggesting ranavirus is not threatening native Chilean species yet. Phylogenetic analysis of partial ranavirus sequences showed 100% similarity with FV3. Our results show a restricted range of ranavirus infection in central Chile, coinciding with X. laevis presence, and suggest that FV3 may have entered the country through infected X. laevis, which appears to act as a competent reservoir host, and may contribute to the spread the virus locally as it invades new areas, and globally through the pet trade.

Introduction

Globally, amphibians are facing an extinction crisis with no precedent, with around 50% of species threatened by extinction (González-del Pliego et al., 2019; IUCN, 2020). Emerging infectious diseases (EIDs) have been increasingly recognized as a threat to biodiversity (Daszak, Cunningham & Hyatt, 2000), and particularly to amphibians (Daszak, Cunningham & Hyatt, 2003; Scheele et al., 2019). In amphibians, two EIDs are well known for their impacts at the population level: chytridiomycosis and ranavirosis. Ranavirosis, caused by infection with a virus in the genus Ranavirus, has been associated with amphibian mass mortality events in North America and Europe (e.g., Green, Converse & Schrader, 2002; Muths et al., 2006; Kik et al., 2011; Cunningham, 2018) and with population declines in the United Kingdom and Spain (Teacher, Cunningham & Garner, 2010; Price et al., 2014; Rosa et al., 2017). Considering the impacts amphibian ranavirosis may have on native amphibian populations worldwide, it is now listed by the OIE as an internationally notifiable disease (Schloegel et al., 2010).

Ranaviruses are pathogens in the family Iridoviridae that are known to cause disease in amphibians, reptiles and fish in many regions of the world (Duffus et al., 2015). Frog virus 3 (FV3), which is the type species of the genus Ranavirus (Chinchar et al., 2009), has been detected in all continents where amphibians are found (Duffus et al., 2015). While most reports on ranavirus-associated outbreaks and mass mortalities come from North America and Europe (Gray, Miller & Hoverman, 2009; Duffus & Cunningham, 2010) and an increasing number of die-offs have been reported in China (e.g., Xu et al., 2010; Geng et al., 2011), ranaviruses remain largely understudied in most regions of the world (Brunner et al., 2021). For instance, few studies have searched for evidence of ranavirus infection in amphibians from South America. FV3-like viruses have been reported infecting wild amphibians from Venezuela (Zupanovic et al., 1998), and causing disease and mortality in wild amphibians from Argentina, Chile, Peru and Brazil (Fox et al., 2006; Soto-Azat et al., 2016; Warne et al., 2016; Ruggeri et al., 2019), and in farmed North American bullfrogs (Lithobates catesbeianus) in Uruguay and Brazil (Galli et al., 2006; Mazzoni et al., 2009). To our knowledge, the presence of ranavirus has not been reported in South American wild or farmed fish or reptiles. In Chile, FV3 infection was first described in 2016 in wild anurans from two populations near the capital city of Santiago in central Chile, involving seven individuals of the invasive African clawed frog (Xenopus laevis), and one individual of the sympatric giant Chilean frog (Calyptocephalella gayi; Soto-Azat et al., 2016). Currently, no ranavirus lineage is known to be endemic to Chile, and available data suggests that the virus may not be native to the country (Soto-Azat et al., 2016). Chile is known for its high rate of amphibian and fish endemism, with most endemic species being threatened and restricted to small areas (Soto-Azat et al., 2015), which makes them prone to the negative impacts of stochastic events such as the emergence of a disease such as ranavirosis. However, the epidemiology of ranavirus in Chile has not been studied; for instance, the origin, extent and impacts of ranavirus are unknown.

Inter-class transmission between amphibians and fish is known to occur. For instance, Brenes et al. (2014) demonstrated that infected mosquito fish (Gambusia affinis) are capable of transmitting a FV3-like virus to naïve Cope’s gray treefrogs (Hyla chrysoscelis) under laboratory conditions. In addition, Moody & Owens (1994) and Ariel & Owens (1997) reported that the amphibian ranavirus BIV produces disease in barramundi (Lites calcarifer) and tilapia (Oreochromis mossambicus) following experimental infections. The ability of ranaviruses to cross species and class barriers has also been observed in the wild. Mao et al. (1999) found identical ranavirus isolates (FV3) from a wild dying threespine stickleback fish (Gasterosteus aculeatus) and a sympatric northern red-legged frog (Rana aurora) from Redwood Creek, California. However, the ability of transmitting the virus depends on the fish species involved (Brenes et al., 2014).

Invasive ectothermic vertebrates may play an important role in the epidemiology of ranaviruses, for instance acting as reservoirs and spreaders (e.g., Robert et al., 2005; Mazzoni et al., 2009; Brunner et al., 2015; Soto-Azat et al., 2016). In addition, humans are known to play a key role in ranavirus spread through translocations of infectious materials or infected individuals (Price et al., 2016). Therefore, pathogen surveillance including native and invasive species of different ectotherm classes are needed to help understand the possible impacts of invasive species on ranavirus spread.

Initially introduced into Santiago in the 1970s, X. laevis is currently widespread in central and north Chile, predominantly inhabiting natural and artificial lentic waters (Mora et al., 2019). It is the only invasive amphibian species known to occur in the country (Veloso & Navarro, 1988; Lobos & Jaksic, 2005). In contrast, there are 26 invasive fish species in Chile, including the widely distributed rainbow trout (Oncorhynchus mykiss), Eastern mosquito fish (Gambusia holbrooki) and common carp (Cyprinus carpio), which have become established in multiple freshwater ecosystems throughout the country (Iriarte, Lobos & Jaksic, 2005).

Here, we hypothesize that ranaviruses infect different species of wild and farmed amphibians and wild freshwater fish from different regions of Chile, and that the African clawed frog (X. laevis) and different species of invasive freshwater fish act as ranavirus reservoirs and spreaders. To increase the knowledge on ranavirus presence, prevalence and geographical distribution in amphibians and invasive freshwater fish from Chile, and to assess the role of invasive amphibians and freshwater fish as reservoirs for ranavirus, we conducted a ranavirus surveillance study of both native and invasive wild amphibians and fish, as well as farmed amphibians, across a large latitudinal range in Chile. We used molecular analysis of tissue samples and buccal swabs through qPCR, and genetic characterization of detected ranavirus.

Materials & Methods

Study area

During the period 2015–2017, wild and farmed amphibians and wild fish were surveyed for ranavirus infection at 19 different sites across Chile over a large latitudinal range (Fig. 1). Sites included lentic and lotic water bodies: natural and artificial ponds, dams, small rivers and streams located within three different ecoregions: Central Andean dry puna, Chilean matorral, and Valdivian temperate forests. We chose the study sites for wild animals considering the concurrent presence of at least one native species and one invasive species. In the case of farmed amphibians, we sampled preserved carcasses provided by a Calyptocephalella gayi ranaculture facility located in Santiago. No other amphibian species is held at this facility.

Figure 1 Map of Chile showing sampling sites for Ranavirus detection from native and invasive amphibians and fish.

The ellipse shows Xenopus laevis current distribution. The asterisk indicates Santiago, the capital city of Chile.

Ethical statement

All applicable institutional and/or national guidelines for the care and use of animals were followed. Procedures were reviewed and approved by the Zoological Society of London (Ref Code WLE718) and Universidad Andres Bello (UNAB13/2015) bioethics committees. Amphibian and fish collections were carried out under permits granted by the Chilean Agricultural and Livestock Service (SAG; N° 212/2016, 1334/2017 and 2793/2017) and the Chilean Fishery and Aquaculture Subsecretary (SERNAPESCA; N° 2036/2015).

Sampling

We collected samples for ranavirus detection during September–December in Northern Chile, and during the Austral summer (December–March) in Central and Southern Chile, coinciding with the warmer months and the breeding season at all sites. All individuals were collected during day sessions, and each site was visited only once. We established 57 as the minimum number of individuals to be collected at each site in order to detect at least one positive individual if the virus was present. We considered a 95% confidence interval, and assumed 100% sensitivity of the ranavirus detection molecular techniques and a previous ranavirus prevalence of 4.3% for amphibians in central Chile (Soto-Azat et al., 2016). For most species, sampling stopped either when the minimum required number of individuals from each species present at a site was achieved, or at the end of the day. In the case of Xenopus laevis, because of Chilean regulations, all individuals captured during the sampling session were euthanized and analyzed for ranavirus presence.

We sampled tadpoles of two abundant, widely distributed native amphibian species: the Andean spiny toad (Rhinella spinulosa) and the four-eyed frog (Pleurodema thaul), as well as adults of the invasive Xenopus laevis for ranavirus detection. In addition, we collected and sampled five Calyptocephalella gayi tadpoles found dead in the wild, and ethanol preserved carcasses of 58 recently metamorphosed C. gayi from a mortality event that occurred in a ranaculture facility in Santiago in 2015 were provided to us for ranavirus detection (Table 1). We also sampled two native fish species: galaxia (Galaxias maculatus) and pochas (Cheirodon galusdae), as well as the invasive Gambusia holbrooki, Oncorhynchus mykiss and Cyprinus carpio (Table S1). All the above were considered as the target species of this study, and were euthanized for tissue sampling as decribed below. In addition, when found, we captured adults of R. spinulosa and P. thaul that were present at sampling sites, plus individuals from other native amphibian species found in sympatry with the target species (Table S2). These individuals were non-invasively sampled and released as described below.

Table 1 Summary of tissue samples (liver, kidney and spleen) collected from wild and farmed amphibians in Chile.

Ranavirus presence was tested through real time PCR. Results are presented by study sites (from north to south) and host species. Prevalence is reported as a percent.

Eco-region	Site	Lat	Long	Species	Developmental stage	n	Rv +	Observed prevalence	95% CI	
Central Andean dry puna	Putre	−18.195503	−69.568573	Rhinella spinulosa	Tadpoles	54	0	0	0–6.6	
Central Andean dry puna	Alcolcha	−21.027084	−68.450103	Rhinella spinulosa	Tadpoles	32	0	0	0–10.7	
	Valle de Jere	−23.187166	−67.991194	Rhinella spinulosa	Tadpoles	5	0	0	0–43.4	
	Calama	−22.748641	−68.071030	Rhinella spinulosa	Tadpoles	7	0	0	0–35.4	
Chilean matorral	Rio Elqui	−29.897250	−71.244583	Pleurodema thaul	Tadpoles	81	0	0	0–4.5	
	Ovalle	−30.644806	−71.568204	Xenopus laevis	Adults	48	1	2.1	0.1–10.9	
	Illapel	−31.619583	−71.141833	Pleurodema thaul	Tadpoles	23	0	0	0–14.3	
				Calyptocephalella gayi	Tadpoles	2	0	0	0–65.8	
Chilean matorral	Jardin Botanico	−32.039694	−71.498111	Xenopus laevis	Adults	39	0	0	0–9	
	Villa Alemana	−33.036251	−71.370742	Xenopus laevis	Adults	285	1	0.4	0–2	
		−34.185358	−70.799575	Pleurodema thaul	Tadpoles	58	0	0	0–6.2	
Chilean matorral	Ranaculture facility	−33.385197	−71.645147	Calyptocephalella gayi	Recently metamorphosed	58	0	0	0–6.2	
	aTalagante	−33.686389	−70.908333	Xenopus laevis	Adults	211	1	0.5	0–2.6	
	aRinconada de Maipu	−33.496111	−70.829722	Xenopus laevis	Adults	297	6	2.0	0.9–4.3	
Chilean matorral	Rancagua	−34.185358	−70.799575	Pleurodema thaul	Tadpoles	59	0	0	0–6.1	
		−34.168433	−70.854356	Xenopus laevis	Adults	60	0	0	0–6	
Chilean matorral	Hualañé	−34.974352	−71.850175	Xenopus laevis	Adults	71	0	0	0–5.1	
				Calyptocephalella gayi	Tadpoles	3	0	0	0–56.1	
	Río Mataquito	−35.045768	−71.737533	Pleurodema thaul	Tadpoles	57	0	0	0–6.3	
Valdivian temperate forests	Valdivia	−39.872720	−73.160637	Pleurodema thaul	Tadpoles	57	0	0	0–6.3	
				Total		1,507	9			
Notes.

a Sites were ranavirus was detected before.

For amphibians, we used nets to collect tadpoles of R. spinulosa, P. thaul, and dead C. gayi tadpoles. We collected adults of X. laevis either by hand nets or using chicken liver baited funnel traps. When traps were used, we set them late in the afternoon and checked early the next morning. We collected adults from native species by hand, and these specimens were released at the same site where captured immediately after sampling. For fish, we used hand nets to collect adult Galaxia maculatus, Cheirodon galusdae, Gambusia holbrooki and Cyprinus carpio. We collected juvenile Oncorhynchus mykiss using hand nets or fishing rods (Table S2). Each captured individual was handled with a new pair of vinyl gloves. To minimize any contamination of samples or the spread of pathogens within and between sites, a strict field sampling and disinfection protocol was followed, with reference to Phillott et al. (2010).

According to Gray, Miller & Hoverman (2012) and Goodman, Miller & Ararso (2013), analysis of tissue samples increases the probability of detecting ranaviruses compared to the analysis of non-invasively acquired samples. Therefore, to minimize the impact on native amphibian populations, we only collected tadpoles for tissue sampling. In the case of X. laevis, only adults were collected, as we did not find tadpoles at the study sites where this species was present.

Collected amphibian tadpoles, adult X. laevis and fish were euthanized at their capture sites using an overdose of the anaesthetic tricaine methane sulfonate (Dolical 80%, Centrovet), buffered (pH 7) with sodium bicarbonate (Bayley, Hill & Feist, 2013). Calyptocephalella gayi carcasses from the ranaculture facility were rinsed with distilled water before necropsy. Gross examination of amphibian and fish viscera were conducted by a veterinarian following Miller, Gray & Storfer (2011). Most histopathological changes associated with ranavirus occur in liver, kidney and spleen (Miller et al., 2015). Thus, for ranavirus detection, we obtained samples of these three organs and placed them in individual vials containing 95% ethanol. In addition, we obtained non-invasive oral swab samples from all adult native amphibians captured (see Table S2), by rotating a sterile rayon tipped swab (Medical Wire) for 3–5 s against the buccal mucosa (Gray, Miller & Hoverman, 2012; Goodman, Miller & Ararso, 2013); swab tips were stored in 1.5 ml sterile vials containing 95% ethanol prior to nucleic acid extraction.

Ranavirus qPCR

We extracted genomic DNA from pooled, homogenized tissue (liver, kidney and spleen) from each individual, using the DNeasy blood & tissue kit (Qiagen, Hilden, Germany) following the manufacturer’s protocol. Extraction controls (all reagents, but no tissue) were included in each extraction batch to test for cross contamination. All samples were tested for ranavirus presence/absence using a specific qPCR assay established by Leung et al. (2017), which targets a 97bp region of the MCP gene of amphibian associated ranaviruses. This assay has a 100% comparative sensitivity and specificity relative to the most commonly employed end point PCR used as the comparator (Mao, Hedrick & Chinchar, 1997). Briefly, samples were run in duplicate in 20 µl qPCR reactions containing: 10 µl Taqman Universal 2X Master Mix (Thermo Fisher Scientific, Waltham, MA, USA), 5.95 µl nuclease-free water, 1 µl of 10 µM of forward (GTCCTTTAACACGGCATACCT) and 1 µl of 10 µM reverse (ATCGCTGGTGTTGCCTATC) primers, 0.05 µl of 100 µM VIC-labelled probe (TTATAGTAGCCTRTGCGCTTGGCC), and 2 µl of template. qPCR reactions were set in 96 well-plates and run on a StepOnePlus (Applied Biosystems, Waltham, MA, USA) machine along with a no-template control (nuclease-free water), a positive control, and ten-fold serial dilutions of DNA extracted from a cultured ranavirus isolate, RUK13 (Cunningham, 2001), with known viral concentration used as standards (3, 30,300 and 3000 viral copies/2 µl). The limit of detection for this assay is 4.23 MCP copies per reaction (95% detection rate), below this value, detection rate falls and there will be some disagreement between replicates (Leung et al., 2017). Samples were considered positive only if sigmoidal amplification occurred in both replicates, the CT values fell within the range covered by the standards, and all no template controls were negative. We quantified viral copies per reaction from positive samples using the standard curve, and reported the mean quantity obtained from both replicates. To determine if negative amplifications were due to PCR inhibition, an internal positive control (IPC) was amplified in a subset of 200 randomly chosen samples that tested negative for ranavirus. This was achieved by using a separate qPCR targeting an ultra-conserved non-coding element of vertebrates, described by Leung et al. (2017). We included samples from each species considered in this study. qPCR setup was the same as described above for the MCP gene, and primers and probe were as follows: forward primer (ATGCTGCAATTCAAACTGTCAG), reverse primer (CAGTAAGCAAAATKGGGAAGAAGC) and FAM-labelled probe (CACTGGTTTGCTCAGGGATA), as outlined by Leung et al. (2017).

In addition to the samples collected for the current study, extracted DNA from ranavirus positive samples from three Xenopus laevis frogs and a single Calyptocephalella gayi frog from a previous study (Soto-Azat et al., 2016) were analyzed using qPCR for viral load quantification.

Ranavirus characterization

We analyzed extracted DNA from ranavirus positive samples obtained from this study and from a previous survey (Soto-Azat et al., 2016) for whole genome sequencing. We prepared sequencing libraries using Agilent’s SureSelectXT2 Target Enrichment System for Illumina Paired-End Multiplexed Sequencing, following the manufacturer’s protocol for 100 ng DNA samples, and following all quality control steps. Libraries were pooled and run on a MiSeq System (Illumina, San Diego, CA, USA). We removed the adapters, and then performed sequence quality control and trimming using Prinseq version 0.20.4 (Schmieder & Edwards, 2011). Bases with a quality score lower than 20 were trimmed from both ends of the reads, and reads with a minimum length of 150 bp and maximum length of 225 bp and a mean quality score of 30 were selected. Trimmed and filtered sequences were aligned using Bowtie 2 v.2.3.4.2 (Langmead & Salzberg, 2012). As previously reported in Soto-Azat et al. (2016), the partial sequences obtained from 3 Xenopus laevis and 1 Calyptocephalella gayi showed 100% homology with each other and therefore, we assembled them together for sequencing. The assembled sequence was mapped against 12 published ranavirus genomes obtained from the NCBI nucleotide database, using Bowtie 2. A maximum likelihood tree was built using Mega-X (Kumar et al., 2018).

Prevalence estimation

Ranavirus prevalence within each study site was calculated for each sampled species and for each study site using epi.prev function in epiR package, R version 3.4.3 (Stevenson et al., 2020). Sensitivity and specificity were set as 100%. Confidence interval was set at 95% using the Wilson method.

Results

Sampling

In total, 2,248 individual animals were sampled for ranavirus detection across a latitudinal gradient of 2,500 km across Chile. These comprised tissue samples from 1,507 amphibians (wild tadpoles and farmed metamorphs) from native species and adult X. laevis and 496 fish, and buccal swabs from 245 adult native amphibians. Details of sampled amphibians and fish are shown in Table 1, Tables S1 and S2. No clinical signs or macroscopic lesions consistent with ranaviral disease were observed in any of the sampled individuals.

Ranavirus qPCR

Quantitative PCR for ranavirus infection resulted in nine positive samples, all of which were from the invasive species Xenopus laevis. This result represents 0.5% (9/1752) of ranavirus prevalence in Chile, and 0.9% (9/1011) of the sampled X. laevis. Ranavirus was detected in four of seven sites invaded by X. laevis: Ovalle (2,1%; 1/48), Villa Alemana (0.4%; 1/285), Talagante (0.5%; 1/211) and Rinconada de Maipú (2%; 6/297) (see Table 1). None of the fish sampled and none of the 58 Calyptocephalella gayi frogs obtained from the ranaculture facility tested positive for ranavirus. Our results showed ranavirus is still present in X. laevis populations from Talagante and Rinconada de Maipú were it was reported before (Soto-Azat et al., 2016), and its spread to two sites, Ovalle and Villa Alemana, located northern to the previously known distribution. Here, we present a map with the extant distribution for ranavirus in Chile (Fig. 1), with all positive sites occurring in central Chile within the Chilean matorral ecoregion.

Viral loads from the nine ranavirus positive X. laevis individuals sampled for this study were very low, all being below 10 viral copies per reaction (Table 2). All four ranavirus-positive samples from the previous study yielded enough DNA for sequencing: the three samples belonging to X. laevis individuals ranged from 172 to 662 viral copies per reaction, and the one sample belonging to the native C. gayi had 3,146 viral copies per reaction (Table 2).

Table 2 Ranavirus viral loads in twelve positive African clawed frogs and one Giant Chilean frog from central Chile.

Viral loads are presented as number of viral copies per reaction.

Region	Study site	Sample ID	Species	# Viral copies per reaction	
Chilean matorral	Ovalle	MUR12/17	Xenopus laevis	3.01	
Chilean matorral	Villa Alemana	VA20/16	Xenopus laevis	8.11	
Chilean matorral	Talagante	ETA29/17	Xenopus laevis	3.52	
Chilean matorral	Rinconada	ERI36/17	Xenopus laevis	2.71	
Chilean matorral	Rinconada	ERI49/17	Xenopus laevis	4.06	
Chilean matorral	Rinconada	MRI04/17	Xenopus laevis	2.61	
Chilean matorral	Rinconada	MRI13/17	Xenopus laevis	8.70	
Chilean matorral	Rinconada	MRI15/17	Xenopus laevis	6.57	
Chilean matorral	Rinconada	MRI27/17	Xenopus laevis	4.61	
Chilean matorral	Talagante	RV75a	Xenopus laevis	286.54	
Chilean matorral	Talagante	RV77a	Xenopus laevis	662.28	
Chilean matorral	Talagante	RV78a	Calyptocephalella gayi	3,146.30	
Chilean matorral	Talagante	RV82a	Xenopus laevis	172.72	
Notes.

a Samples from the previous study.

Ranavirus characterization

No ranavirus DNA sequences were retrieved from the ranavirus-positive samples collected for this study, and only partial ranavirus sequences were obtained from the four DNA extracts from the previous study. A 531 bp contig was obtained from assembled sequences (GenBank accession number ON788001; see Supplementary Files). This sequence was then aligned against 12 published ranavirus genomes and showed 100% similarity with different isolates of FV3 and FV3-like viruses from North America and the United Kingdom: FV3 (AY548484) isolated from Lithobates pipiens in the USA (Tan et al., 2004), FV3 SSME (KJ175144) isolated from Lithobates pipiens in the USA (Morrison et al., 2014), and RUK13 (KJ538546) isolated from Rana temporaria in the UK (Price, 2013). See Fig. 2 for a comparative phylogeny including the Chilean ranavirus sequence.

Figure 2 Maximum likelihood phylogenetic tree showing the evolutionary relation between the ranavirus major capsid protein (MCP) gene partial sequence (531bp) obtained from wild amphibians from central Chile (RV Chile) and a panel of 12 ranaviruses genomes.

Ranavirus genomes were downloaded from GenBank. We used the bootstrap method, with 100 bootstrap replications. Numbers at nodes indicate bootstrap support. RV Chile was grouped in the FV3 clade, showing 100% similarity with FV3, RUK13 and FV3 SSME. Short-finned eel ranavirus (SERV) was set as an outgroup.

Discussion

Our results show that ranavirus is widely distributed at low prevalences in central Chile, appearing to be associated with Xenopus laevis presence. We only found evidence of ranavirus infection in X. laevis, despite numerous samples from other amphibian and fish species being examined from disparate areas of Chile, including specimens of Calyptocephalella gayi, a species in which ranavirus was detected before (Soto-Azat et al., 2016). Also, our results show the virus is still present in the two sites near Santiago were it was detected in 2011: Talagante and Rinconada de Maipú (Soto-Azat et al., 2016). We also detected ranavirus infecting X. laevis from two new sites: Villa Alemana and Ovalle, extending ranavirus distribution to the north (Fig. 1). This information suggests X. laevis is contributing to ranavirus persistence and spread in the country.

All ranavirus-positive X. laevis detected in this study were apparently healthy individuals. Covert FV3 infection by immunocompetent individuals, with development of clinical disease only in immunosuppressed animals, has previously been reported for adult X. laevis raised in captivity (Robert et al., 2005; Robert et al., 2007). Both immunocompetent and immunosuppressed frogs are capable of transmitting the virus to susceptible hosts (Robert et al., 2007). In the current study, the PCR-positive X. laevis had low viral loads and it is possible that in other individuals virus might have been present but below detectable limits (4.23 viral copies per reaction, Leung et al., 2017). Therefore, the infection rate for ranavirus in X. laevis presented here may be underestimated. Low viral loads detected in infected individuals might be associated with the beginning or end of infection, considering the ability of adult X. laevis to clear infection after a second exposure (Gantress et al., 2003). A seasonal pattern of ranavirus outbreaks has been reported, with most of them occurring during the warmer months (Brunner et al., 2015). Higher viral loads detected in X. laevis from the previous study (Soto-Azat et al., 2016) may be associated with a first exposure to the virus, or with the time of sampling (two individuals were collected during summer and two during autumn). However, long-term monitoring of infected populations is needed to confirm this hypotheses.

Non-detection of ranavirus in tissue from native amphibian tadpoles included in this study may be related with a potentially low susceptibility of the target species. In addition, native amphibians (Pleurodema thaul tadpoles) were found sharing the same aquatic site as adult X. laevis only in one of the four positive sites: Villa Alemana. No adults from this or any other native species were seen at this site, which may be associated with diurnal sampling. In the remaining positive sites, native amphibians were found at nearby water bodies, lowering the chances of transmission by direct contact between infected and susceptible individuals. However, considering the ability of X. laevis to migrate through water canals and also through land (Lobos & Jaksic, 2005), transmission to sympatric native amphibians may occur. Even though we analyzed samples (buccal swabs) of several threatened native species (see Table S1), none of them tested positive for ranavirus infection. Our results suggest that ranavirus may not be present yet, or may be occurring at low prevalence or below detectable limits in the sampled native amphibian species. Buccal swabbing is known to have lower sensitivity (∼65% sensitivity) when compared with ranavirus detection in internal tissue, leading to false negatives (Gray, Miller & Hoverman, 2012; Allender et al., 2013; Goodman, Miller & Ararso, 2013). In addition, higher susceptibility of tadpoles of some species to ranavirus infection has been reported (Hoverman, Gray & Miller, 2010; Hoverman et al., 2011); therefore, results should be interpreted cautiously. As the effects of ranavirus infection on endemic threatened species are unknown, active disease surveillance and population monitoring of native amphibians should be focused on water bodies recently invaded by X. laevis and in those where invasion of X. laevis is imminent. All C. gayi carcasses from the ranaculture facility tested negative to ranavirus infection and mortality was determined to be caused by a chytridiomycosis outbreak (Alvarado-Rybak et al., 2021). To our knowledge, no ranavirus associated mass amphibian die-offs have been reported in Chile to date; however, the possibility of unnoticed or future mortality events due to ranavirus cannot be discarded. Ranavirus challenge experiments using the “Chilean lineage” in different native amphibian species and life-stages are recommended to further assess species susceptibility and possible impacts of ranavirosis to local amphibians.

Xenopus laevis is known to be a successful invasive species (Ihlow et al., 2016; Measey et al., 2012; Mora et al., 2019). It is known to have a preference for a Mediterranean climate which occurs in central Chile, within the Chilean matorral eco-region (Lobos & Measey, 2002), although recent findings suggests the species is adapting to different environmental conditions (Hill, Lawson & Tuckett, 2017). Habitat suitability analyses have shown a high potential for X. laevis to colonize new areas of Chile and neighboring countries (Barbosa, Both & Araújo, 2017; Ihlow et al., 2016). According to Bielby et al. (2020), a single host species can be responsible for maintaining ranavirus within a community; our results point to X. laevis as the reservoir host of ranavirus at all sites in Chile where the virus has been detected. The continued range expansion of X. laevis in Chile (Mora et al., 2019), therefore, may result in further spread of ranavirus in this country.

Even though fish may act as reservoirs for ranavirus (Gray, Miller & Hoverman, 2009; Brenes et al., 2014), no ranavirus DNA was detected from any of the sampled fish. However, the role of fish as ranavirus reservoirs may depend on the ranavirus species and the host species (Jancovich et al., 2001). Laboratory trials involving Cope’s gray treefrog tadpoles (Hyla chrysoscelis) and mosquito fish (Gambusia affinis), have shown that, while some species of fish can become infected with FV3, transmission from fish to frogs is low (Brenes et al., 2014). The same experiment showed that H. chrysoscelis tadpoles were not capable of transmitting FV3 to mosquito fish. In areas with ranavirus positive X. laevis, we did not detect ranavirus DNA in samples from fish.

While whole genome sequencing of viral DNA obtained directly from infected tissues is possible (Depledge et al., 2011), most (>99.8% or more) of the obtained reads are of the host DNA. Therefore, when low numbers of viral copies are present in the sample, viral genome coverage is expected to be low (Houldcroft, Beale & Breuer, 2017). It is likely that only partial sequences of ranavirus DNA were obtained in the current study because only low numbers of viral copies were present in the examined tissues.

Phylogenetic analysis of the four ranavirus DNA sequences obtained from Chilean amphibians identified a single genotype which grouped within the FV3 clade. The viral sequence from the only known case of a ranavirus infecting a native Chilean amphibian (C. gayi) was identical to that of the virus infecting X. laevis (Soto-Azat et al., 2016). This C. gayi individual was found dead in 2011 with signs of ranavirosis in a pond from Talagante that also contained X. laevis that tested positive for ranavirus (Soto-Azat et al., 2016). Five years later, our results confirm ranavirus is still present in X. laevis in this system; however, native species were not found during sampling. These results suggest that ranavirus infects mainly X. laevis, and an apparent absence of the virus outside the distribution range of this invasive species in the country (see Fig. 1) suggests a recent introduction of ranavirus to the country, likely associated with the introduction of X. laevis. This further supports the hypothesis that X. laevis can act as a reservoir for ranavirus (Robert et al., 2007), and as such this amphibian could be contributing to the spread of ranavirus globally via international trade for pet and scientific purposes (Weldon et al., 2004; Fisher & Garner, 2007; Soto-Azat et al., 2010; Van Sittert & Measey, 2016), and locally through invasive population dynamics (Measey et al., 2012; Soto-Azat et al., 2016).

Frog virus 3 was first isolated from Rana pipiens in North America in the early 1960s (Granoff, Came & Rafferty, 1965). Since then, closely-related viruses have been detected in all continents where amphibians exist (Duffus et al., 2015). RUK13, an FV3-like virus, was first isolated from a common frog (Rana temporaria) in the UK in 1995, probably following an incursion from North America through the pet trade (Hyatt et al., 2000; Cunningham, 2001; Price, 2013; Price et al., 2016). In Chile, the initial introduction of X. laevis into the wild is thought to have occurred in the 1970s via escaped or deliberately released animals intended to be used in research (Lobos & Jaksic, 2005; Mora et al., 2019). Although X. laevis is currently imported to Chile from USA (e.g., Lee-Liu et al., 2014; Lee-Liu et al., 2018), the origin of the specimens imported in the 1970s is unknown (Lobos et al., 2014). Lobos et al. (2014) established that invasive X. laevis in Chile have low genetic diversity, suggesting that the invasion derived from a single introduction event, and available data suggest that the specimens may have been imported from USA or UK suppliers (Lobos et al., 2014; Nace, Waage & Richards, 1971; Van Sittert & Measey, 2016). Considering FV3-like viruses have been estimated as being introduced into the UK in the late1980s (Cunningham, 2001), and the introduction of X. laevis to Chile occurred in the 1970s, it may be more likely that FV3 was introduced to the country along with X. laevis imported from the USA. Recently, Measey (2017) reported that the majority of wild caught X. laevis imported to the USA come from Chile. He also stated that 99.6% imported individuals are moved for the pet trade. This information is of extreme concern, considering traded individuals could be carrying FV3 and also the chytrid fungus Batrachochytrium dendrobatidis (Solís et al., 2010; Soto-Azat et al., 2016), and that infected individuals could have been exported to other countries besides USA. Releases of potentially infected individuals into the wild may have negative impacts on native amphibian populations in the USA and elsewhere (Measey, 2017). Further research concerning movement of wild X. laevis between Chile, USA, UK and other countries could help track ranavirus spreading via trade. Here, we emphasize the importance of the establishment and reinforcement of importation regulations, sanitary controls and border controls to reduce the probability of the introduction of alien pathogens such as ranavirus and other known and unknown pathogens along with imported wild animals.

Conclusions

Our results suggest that Xenopus laevis may be a competent reservoir for FV3-like ranavirus. Its epidemiological role in Chile could include the maintenance of the virus in the environment and the spread to new sites as this alien amphibian invades new areas. We were able to detect ranavirus only in central Chile, and only in association with X. laevis; however, further sampling efforts in north and south Chile are needed to confirm absence of this emerging pathogen in other parts of the country. Low infection burdens and the absence of clinical signs found in all ranavirus-positive X. laevis are consistent with its previously-reported resistance to ranavirosis. Higher viral loads and lesions compatible with ranavirosis found in the only positive native amphibian, a Calyptocephalella gayi frog, suggests that this species may have higher susceptibility to ranavirus infection and disease than other species native to Chile; however, further investigation is needed to conclude this. We did not find evidence supporting an active role of invasive freshwater fish in ranavirus epidemiology in Chile. Active surveillance of amphibian sites within the range of X. laevis should be established to detect incidents of ranavirosis in native species, and concurrent longitudinal population monitoring conducted to determine any negative impact at the population level. Overall, our results encourage the establishment of X. laevis control strategies, highlighting the importance of avoiding the spread of this species to ponds and streams inhabited by native amphibians within Chile, and the export of wild caught individuals to other countries.

Supplemental Information

Table S1 Summary of tissue samples (liver, kidney and spleen) collected from wild freshwater fish in Chile

Ranavirus presence was tested through real time PCR. Results are presented by host species and study sites.

Click here for additional data file.

Table S2 Summary of buccal swabs obtained from wild adult native amphibians in Chile

Ranavirus presence was detected through real time PCR. Results are presented by host species and study sites.

* Vulnerable. ** Endangered. *** Critically endangered.

Click here for additional data file.

File S1 Commands used in the analysis of sequencing data

Click here for additional data file.

File S2 Chilean Ranavirus (RV CHILE) Partial Sequence

Click here for additional data file.

We thank René Monsalve, Saulo Lebuy, Marta Mora, César Cuevas and Diego Peñaloza for their important fieldwork support. We also thank Jürgen Rottman and Parque Safari for allowing sampling within their facilities.

Additional Information and Declarations

Competing Interests

Author Contributions

Animal Ethics

Field Study Permissions

DNA Deposition

Data Availability

The authors declare there are no competing interests.

Alexandra Peñafiel-Ricaurte conceived and designed the experiments, performed the experiments, analyzed the data, prepared figures and/or tables, authored or reviewed drafts of the article, and approved the final draft.

Stephen J. Price performed the experiments, analyzed the data, authored or reviewed drafts of the article, and approved the final draft.

William T.M. Leung performed the experiments, analyzed the data, authored or reviewed drafts of the article, and approved the final draft.

Mario Alvarado-Rybak performed the experiments, authored or reviewed drafts of the article, and approved the final draft.

Andrés Espinoza-Zambrano performed the experiments, authored or reviewed drafts of the article, and approved the final draft.

Catalina Valdivia analyzed the data, authored or reviewed drafts of the article, and approved the final draft.

Andrew A. Cunningham conceived and designed the experiments, authored or reviewed drafts of the article, and approved the final draft.

Claudio Azat conceived and designed the experiments, authored or reviewed drafts of the article, and approved the final draft.

The following information was supplied relating to ethical approvals (i.e., approving body and any reference numbers):

Zoological Society of London (Ref Code WLE718) and Universidad Andres Bello (UNAB13/2015) bioethics committees provided full approval for this research.

The following information was supplied relating to field study approvals (i.e., approving body and any reference numbers):

Amphibian and fish collections were carried out under permits granted by the Chilean Agricultural and Livestock Service (SAG) and the Chilean Fishery and Aquaculture Subsecretary (SERNAPESCA), respectively (SAG; N° 212/2016, 1334/2017 and 2793/2017), and (SERNAPESCA; N° 2036/2015).

The following information was supplied regarding the deposition of DNA sequences:

The RV CHILE partial sequence is available in the Supplemental Files and at GenBank: ON788001.

The following information was supplied regarding data availability:

The ranavirus partial sequence obtained from infected Chilean amphibians and the commands used in the analysis of sequencing data are available in the Supplemental Files.

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
