# Peer review of "Is Xenopus laevis introduction linked with Ranavirus incursion, persistence and spread in Chile?"

_PeerJ, doi:10.7717/peerj.14497_

## Round 0.1 · original submission · Major Revisions

I have received insightful responses from two reviewers, both of whom suggest that your manuscript may be acceptable for publication given Major Revisions. In addition to your manuscript, I will require a detailed response from you about how you responded to each of the comments of the reviewers, as well as my own directed comments below:

1. Your introduction needs to provide much more specific aims and/or objectives for your study. You currently frame this as a 'look see' surveillance study. This makes the rest of the manuscript very hard to assess, as the reader has little idea of what you wanted to achieve. As mentioned by both reviewers, the introduction also needs more background information relevant to your approach. Some of this is already in the manuscript but needs to be repurposed to the introduction.

2. Methods - I agree with Rev2 that this section could be better organised. In particular, it would be useful to know how your study was planned in relation to numbers of individuals sampled: i.e. did you sample until you met a minimum target, or did you stop sampling after a specific time spent searching?

4. Results - You may want to organise your results by subheadings used in your revised methods. Please pay specific attention to legends in figures and tables.

3. Discussion - in all cases where you found ranavirus in Xenopus, you only sampled tadpoles of other species. You don't discuss this discrepancy, neither do you justify the biased nature of your sampling for larvae of some species and adults of others (contrast L137 with Table 1). This may be legitimate, but needs to be established and/or discussed.

A couple of small points:

L46: 'feral' is a term used for invasive populations of domesticated animals and is inappropriate to Xenopus. I'm guilty of having used this inappropriately in early publications.

L324: Although a preference for Mediterranean climates was originally claimed, I don't see this in the current distribution of invasive populations.

L311: The movement of animals between USA and Chile is interesting. I recently discovered that the movement of wild caught animals from Chile to USA (see Measey Salamandra 53: 398-404) was conducted by XenopusOne and that these animals were then sold to laboratories.

I look forward to your revision and seeing this important work published.

Reviewer 1 ·

Basic reporting

The authors focus in two major issues concerning amphibian conservation: invasive species and pathogens. The study covers an extensive area in Chile, most of which seem to be inhabited by invasive species, and for this effort I commend the authors. Thus, they used different methods for pathogen screening and detection of live and dead animals, they investigated a mass mortality event occurred in a frog farm, and they looked into the genetic aspects of the strain found in Chile, enriching their work. Overall, the manuscript could be better organized and structured to facilitate its reading and understanding. Below you’ll find some important points to be addressed:
Introduction:
- Provide more background to reinforce the importance of investigating ranavirus presence in Chile - ranavirus is a notifiable disease (OIE), panzootic, likely introduced via the live animal trade (such as Xenopus or invasive fish). Provide a clear hypothesis and clearly state the goals of this study. Add more paragraphs if necessary (you only used 2 pages against 6 for discussion).
Methods:
- Methods should be clear: at first, I thought you’d used qPCR for presence/absence of ranavirus in samples collected for this study, and quantification only for those from the previous study
- Add information on which type(s) of aquatic environment you sampled, how many times you visited each site, if sampling occurred during daytime or at night, etc.
- Add references for how you sampled specimens. Clarify that the methods applied were different for live and dead individuals.
- Clarify what captive animals mean and which species can be found in these captivities (are native amphibians farmed in Chile? Or X. laevis? Why didn’t you sample captive fish from aquaculture facilities?).
- The methods for qPCR are not very clear and lots of information seem to be missing. It seems that you used Leung et al. (2017) with modifications, but I am not sure. This is extremely important if someone wishes to reproduce your study. Also, you don’t say what you considered as positive (higher than 3 copies?), what did you do when one well amplified and the other from the same sample did not? Did you run the sample for a third time, did you consider it as negative? Also, how did you calculate loads, did you use the mean load from your duplicates?
Results and Discussion:
- I don’t recall you discussing about ranavirus persistence and spread. I don’t think you state to have found ranavirus (or Xenopus) in new sites or reporting if positive individuals were collected in the same year to imply that ranavirus is persisting over time and spreading
- You hypothesized that Xenopus is responsible for the introduction of ranavirus in Chile, and with your results I think you can argue that (for example, you found 100% similarity between your sequences and FV3 from the USA). This is an interesting result, so explore it more! Use your results to discuss your hypothesis and suggest other hypothesis to be tested in other studies.
- Support for your clade is not very high (49) but you don’t add any discussion about that…
- It’s not clear if the association between ranavirus and X. laevis is a hypothesis you are testing in this study or if this is result/discussion.
- Would you consider the low loads presented in Table 2 the beginning of infection, perhaps? I think it would be interesting to discuss this result as loads are so different from those frogs collected for Soto-Azat et al. (2016)
Conclusion:
- “Our results indicate that X. laevis is a competent reservoir for FV3-like ranavirus”. I don’t think you have enough to conclude that. You can suggest this is true, you can say your results seem to corroborate that this is true. However, your prevalence is too low, you don’t provide details on your sample sites (I don’t know if individuals were collected from the same aquatic habitat, for example)

Experimental design

In addition, there are some minor and major issues that deserve attention before the paper acceptance. For example, I personally like the use active voice to describe what was done in the methods (minor). However, the methods are not described with sufficient detail and information (major), and the goals and hypotheses are not clear stated (major).
Because you analyze data collected for this study AND from the previous study, at some points you reported the results form that study (which have already been published). Results from Soto-Azat et al. (2016) are not results of you study (e.g., sequences 100% compatible with FV3 strain); therefore, they are part of the discussion.
Methods should be more clear.

Validity of the findings

In Figure 1 the eco-regions are present, but not in tables. In fact, in table 1, there is something called "regions" and a few codes with no legend. Be consistent.
In Figure 2 there are some missing information here. For example, are Ranavirus partial sequence (531bp) from which gene, MCP? The “panel of 12 ranaviruses” was downloaded from GenBank, so add this information to the table legend. Numbers at nodes indicate bootstrap support, correct? How many replicates? I don’t think you say that in the methods.
In Table 1, why not presenting the results from oral swabs as well? Again, what does region mean? There is no label/legend for codes (like in Table 2). Maybe you could divide the results into eco-regions, like you did in the map
Authors provide raw data and supplementary material. However, the access number in GenBank doesn't work yet.

Additional comments

The manuscript title “Is Xenopus laevis introduction linked with Ranavirus incursion, persistence and spread in Chile?” is too wide and I don’t think they have evidence to answer such question. I would recommend something more direct, such as “presence of Ranavirus in invasive frog in Chile suggests the species as reservoir for the lethal virus” or “presence of Ranavirus in Chile seems to be associated with Xenopus laevis”

The manuscript is of great importance and deserves to be published. However, it needs to be much improved. The methods are poorly described, I am not sure what the goals were, the discussion is bigger than necessary and missing important points, and there are important references in the methods that should be provided (e.g., Goodman et al. 2013; Miller et al. 2015).
More comments and inquiries can be found in the attached pdf.

Annotated reviews are not available for download in order to protect the identity of reviewers who chose to remain anonymous.

Reviewer 2 ·

Basic reporting

Generally, this manuscript meets all of the 'basic reporting' requirements. However, please see my comments for 'experimental design' component, which is where I explained my concerns.

Experimental design

Peñafiel-Ricaurte et al. report on a field-based study of ranaviruses (RV) in amphibians and fish from several locations across Chile. As the authors point out, RVs are relatively understudied in this region of the globe. Their study detects RV in a handful of non-clinically/sublethally infected Xenopus laevis, an introduced species in Chile, at multiple wetlands. Their study does not find any evidence of RVs infecting any other species although previous studies suggest that RVs are likely infecting multiple species in Chile. Unfortunately, there are several issues in the Methods that make it difficult to ascertain how reliable their findings are. Hopefully these issues can be addressed with better, detailed explanations. However, there are some issues that, unless adequately addressed, makes these data unpublishable because they are unreliable and cannot be interpreted in meaningful ways. I wish these researchers every success in their continued work on this important topic, whatever happens with this particular manuscript.

I’ve separated my comments into Major and Minor comments. I have included all my comments under the 'experimental design' component of the review platform because my major concerns are directly linked to experimental design, and follow over to concerns about 'validity'.

Major Comments:

- lines 81 – 94: This block of text contains relevant information that isn’t organized in a useful way. I would suggest organizing the information along themes and then citing the corresponding articles. For example, a sentence or two about RV detected in introduced wild amphibians (REFS); a sentence or two about RV detected in ranaculture/aquaculture facilities or species (REFS)…. As well, it warrants comment from the authors whether RV have been looked for in fish and reptiles in their region, and if so, what has been found to date, particularly since their own study includes fish.

- line 113 – 119, lines 126 - 134: Information in these two blocks of text appear to be overlapping, with choice of study sites (ostensibly the first block of text, based on subtitle but not content) not well explained. As well, there are species listed in Tables that were apparently lethally sampled but are not mentioned in this section of the Methods. There are also issues with sentence structure, punctuation, and grammar that impede clarity and content.

- Table 1 and related areas of text: I have some inter-related questions that stem from my understanding/personal experiences of RV in wild amphibian populations in temperate North America, where RV outbreaks are highly seasonal, affect different life stages differently, affect different species differently, and usually move through populations in epizootics. My questions are aimed at digging a bit into your results, and hopefully moving them into the ecology of RVs rather than just putting ‘dots on a map’ of where RV was detected.
When were samples collected, specifically? Line 128 indicates a 6 month window across each of 3 years. The 6 month window is very wide in general, and also relative to the life cycles of amphibians. I wonder if there might be unintentional biases in the data because of the timing of sampling. For example, was it too early or too late in the active season to detect RV waves? As well, were all samples from a given species at a site collected at the same time, or do the prevalences and 95% CI reflect pooled animals from multiple sampling trips, potentially across years? Were all species at a site sampled at the same time or were different species sampled at different times. If the latter (ie, pooled across sampling events), estimates of prevalence must be discussed in those contexts because they are not equivalent to estimating prevalence in group of animals that were all sampled at the same time in the same place. RV waves move fast spatially and temporally in most other systems that have been investigated so far, so the probability of detecting a RV+ animal is not uniform across species, life stages, time, or space at the spatial scale of a wetland, let alone a whole landscape as diverse as the one your team sampled.

- Table 1 and related areas of text: how was ‘adult’ versus ‘post metamorph’ life stage determined? Is a ‘post metamorph’ a recently metamorphosed animal just leaving the natal pond for the first time? Similarly, how was it determined if an animal was a sub-adult but not just recently metamorphosed? Some clarification around this is warranted given the well documented differences in outcomes of RV across life stages and ages of anurans in other systems. If it wasn’t possible to determine whether an animal was actually an adult (ie sexually mature), perhaps make a statement to that effect in the Methods and explain the use of terminology.

- line 162: It’s not clear whether your team developed new qPCR primers, or if you followed Leung et al. including using their primers. If you designed your own, considerably more information is required on how you assessed the validity of your primers. If you used the primers described in Leung et al., the information in this part of the text needs to be clarified.

- line 173 or thereabouts: how did you establish whether a sample was declared RV+? (ie threshold?) What did you do for samples when one well indicated RV and the other well did not?
- line 174 or thereabouts: what happened with the positive controls - What were the detection limits observed in your study?

- Table 2 and related text in Methods and Results including line 216:
‘units’ are missing from the viral load values. For example, 3 viral copies per what volume or amt? As well, do the values presented in Table 2 indicate an average viral load across the two wells? Were zeros included in those averages? Some information for your consideration on how to report viral loads can be found in Hoverman et al. 2011 EcoHealth. There are other sources out there as well. Whatever you decide, please report details regarding quantification.

- Line 175: what was the outcome of your Internal Positive Controls? Why were only <10% of the samples assessed for the possible effects of PCR inhibition given how rampant inhibitors are? <10% is pretty low, even factoring in cost considerations. Did the 200 randomly chosen samples happen to encompass all years, all species? If all species, all life stages, all years, etc weren’t captured, a significant and avoidable blind spot was created, especially in light of how few samples were assessed. Setting that aside, how did you handle situations when 1 well amplified but the other did not for the IRC? How often did this happen? This is critical information that must be included in order to interpret the results, particularly since there was <1% prevalence overall, from a single species, and at very low viral loads.

- line 254: detectable viral loads. Again, there are issues with no units associated with the viral load. That said, assuming units are comparable between this statement and data reported in Table 2, several of the RV+ samples are below the detectable viral load. Something is amiss here. Either details are missing or there are issues with how quantities of virus are being reported.


Minor Comments:

- line 56: what is the “previous study” referred to in this sentence? It is out of context in the Abstract. Reader doesn’t learn about this “previous study” until halfway through the manuscript.

- line 59: the results suggest that FV3 isn’t spreading – perhaps it could, but doesn’t appear to have done so at these study locations. I suggest deleting “… spreading the virus as it invades new areas” as there is limited evidence from your study that this has happened.

- lines 67 – 68: I suggest deleting the sentence about Bd, and editing the preceding sentence accordingly. Although Bd is an important amphibian pathogen, it is not relevant to this study about RV in an introduced species of amphibians.

- in several places, it seems that ‘e.g.,’ should be used prior to a list of citations since you are giving examples of studies that support your argument, rather than providing the definitive, singular location of a piece of information or an exhaustive list of studies. For example, lines 78 and 79; elsewhere throughout paper.

- lines 76 – 83, and text immediately following: This block of text is somewhat self-contradictory, perhaps misleading, and a bit out of date with respect to the literature. For example, it doesn’t make sense to say that RV aren’t studied in S America and then proceed to detail several studies of RV from S America… I think this is a matter of word smithing and editing to more cleanly get across your argument that a lot remains unknown about RV in S America, particularly in light of the amphibian biodiversity and the ravages of introduced species in S America. I encourage the authors to consider information presented by Brunner et al. 2021 FACETS and Blaustein et al. 2018 Diversity because there may be helpful ideas for buttressing your arguments about the importance your study.

- line 94-95: This sentence requires a citation as to the source of the statement. As well, there are typos/grammar issues with the sentence.

- Table 1 and related areas of text including line 213. Were the C.gayi from ranaculture facility wild? The title of the table indicates all entries are from wild populations. Some clarification might be needed here.

- line 173: space missing between 30 and 300

- Results, Discussion and Conclusions, generally: Based on the information presented in this paper, it would seem that X.laevis isn’t particularly good at transmitting virus to other species in this region. This is assuming that rampant PCR inhibition isn’t the underlying explanation for low prevalence and low viral loads, and also assuming that there weren’t sampling biases that prevented more widespread detection across space, time, and species (please see comments above). Thus, assuming the issues with the Methods can all be solved with better explanation, rather than reflecting fatally flawed execution, it would seem that much of the extrapolation is overstated. These sections could be considerably shortened and redundancy removed.

- lines 346-348: comments re ranavirosis in C.gayi are beyond the scope of this paper. Presumably this information is already covered in the paper that first reports the findings and therefore comment could potentially be made by citing that paper. But, the data presented in this paper alone do not allow for comment on ranavirosis in any species other than Xenopus because RV was not detected in any other species.

Validity of the findings

Please see my comments for 'experimental design' component.

---

## Round 0.2 · accepted · Accept

Thank you for addressing the reviewer comments. The reviewer makes some comments on ways in which you could improve the readability of the methods section. I think that most of this reads well, but please take the opportunity to polish the text where you can. Thanks for contributing this manuscript to PeerJ.

Reviewer 1 ·

Basic reporting

The new version of the manuscript "Is Xenopus laevis introduction linked with Ranavirus incursion, persistence and spread in Chile?" is much improved. However, as I realised there are native English speaker co-authoring the manuscript, I would recommend them to review the text once again before publication because I found a few grammar mistakes throughout the text (e.g., line 61). I also found some errors such as extra space (e.g.: line 64), missing (e.g.: line 100) or extra (e.g.: line 165) italicized words, some incoherence such as italicized FV3 (L. 46) and non italicized FV3 (L. 104), etc.

Experimental design

Even though the methods are described with sufficient detail and information to replicate, I still find it complicated to read. For example, I would suggest the authors to re-write the "sampling" section. Clarify that the study focused on tadpoles (L. 166-167) but that they also sampled adults found during fieldwork, but that different methods for Rv screening were used - e.g., the sentence "to avoid causing a negative impact on native amphibian populations, we only collected tadpoles for tissue sampling" is not clear (L. 169-171). Did the authors collect tissues from all tadpoles but not from adults and fish? what about Xenopus and the carcasses? Also, it would be great if they could reorganize their thoughts throughout the text: first they explain how specimens were captured, then how they sampled specimens for Rv, and then they go back to how specimens were captured. Paragraph on line175 should come before the previous paragraph (L. 169-174). Maybe lines 197-200 could be moved to the qPCR section and placed with the methods for viral quantification (L. 220). This part (viral quantification) could be a new paragraph

Validity of the findings

Their findings are very important and conclusions well stated.